# Different Approaches to Appraising Systematic Reviews of Digital Interventions for Physical Activity Promotion Using AMSTAR 2 Tool: Cross-Sectional Study

**DOI:** 10.3390/ijerph20064689

**Published:** 2023-03-07

**Authors:** Karina Karolina De Santis, Katja Matthias

**Affiliations:** 1Department of Prevention and Evaluation, Leibniz Institute for Prevention Research and Epidemiology—BIPS, 28359 Bremen, Germany; 2Leibniz Science Campus Digital Public Health Bremen, 28359 Bremen, Germany; 3Faculty of Electrical Engineering and Computer Science, University of Applied Science Stralsund, 18435 Stralsund, Germany

**Keywords:** AMSTAR 2, digital intervention, physical activity, systematic review, appraisal

## Abstract

High-quality systematic reviews (SRs) can strengthen the evidence base for prevention and health promotion. A 16-item AMSTAR 2 tool allows the appraisal of SRs by deriving a confidence rating in their results. In this cross-sectional study, we aimed to assess and compare two approaches to appraising 30 SRs of digital interventions for physical activity (PA) promotion using AMSTAR 2. Approach 1 (appraisals with 2/16 items) was used to identify SRs with critically low confidence ratings. Approach 2 (appraisals with all 16 items) was used (1) to derive the confidence ratings, (2) to identify SR strengths and weaknesses, and (3) to compare SR strengths among subgroups of SRs. The appraisal outcomes were summarized and compared using descriptive statistics. Approach 1 was quick (mean of 5 min/SR) at identifying SRs with critically low confidence ratings. Approach 2 was slower (mean of 20 min/SR), but allowed to identify SR strengths and weaknesses. Approach 2 showed that confidence ratings were low to critically low in 29/30 SRs. More strengths were identified in SRs with review protocols relative to SRs without review protocols and in newer SRs (published after AMSTAR 2 release) relative to older SRs. Only two items on AMSTAR 2 can quickly identify SRs with critical weaknesses. Although most SRs received low to critically low confidence ratings, SRs with review protocols and newer SRs tended to have more strengths. Future SRs require review protocols and better adherence to reporting guidelines to improve the confidence in their results.

## 1. Introduction

High-quality systematic reviews (SRs) can strengthen the evidence base for prevention and health promotion. Although the number of published SRs has increased rapidly over the last 30 years [1], many such SRs of health interventions have weaknesses in their quality [2,3,4] and, thus, may have limited practical use for policy development and health decision-making.

A Measurement Tool to Assess Systematic Reviews, Version 2 (AMSTAR 2) [5] is a tool for appraising SRs of health interventions that was published in late 2017. AMSTAR 2 consists of a questionnaire with 16 items and a comprehensive rating guidance document [5]. The appraisals are conducted by rating 16 aspects of SRs, including research question and review protocol, literature search, study selection and data management, data synthesis, and assessment of potential biases and conflicts of interest. The item ratings are used to derive the overall confidence rating in the results of the SR (critically low, low, moderate or high) [5]. While AMSTAR 2 is an open-access tool with acceptable psychometric properties [5,6,7], the rating time for one SR is approximately 15 to 32 min for experienced users [5,7,8] and could be even longer for less experienced users. Thus, alternative approaches to using AMSTAR 2 for SR appraisals should be tested to potentially reduce the rating time.

In this study, we aimed to assess and compare two approaches to appraising SRs of interventions in public health using AMSTAR 2. Approach 1 (appraisals with 2/16 items) was used to identify SRs with critically low confidence ratings. Approach 2 (appraisals with all 16 items) was used (1) to derive the confidence ratings, (2) to identify SR strengths and weaknesses, and (3) to compare SR strengths among subgroups of SRs.

## 2. Materials and Methods

### 2.1. Protocol and Reporting

This study was performed within our scoping review [9] with a prospectively registered protocol [10]. Except for additional sensitivity analysis, there were no changes between this study and the protocol [10]. The study adheres to ‘The Strengthening the Reporting of Observational Studies in Epidemiology’ (STROBE) guidelines [11]. The STROBE checklist is reported in Appendix A.

### 2.2. Design and Setting

This study used a cross-sectional design to assess appraisals of SRs focusing on evaluation of digital interventions for physical activity (PA) promotion that were published in peer-reviewed journals through March 2021.

### 2.3. Data Sources

Data sources for this study were SRs included in our scoping review [9,10]. These SRs were selected out of 304 reviews of any type that were identified in electronic searches of international databases (Medline, PsycINFO and CINAHL from inception through 19 March 2021) and in bibliographic searches of the included reviews [10]. The inclusion criteria for this study are based on the PICOS framework: (1) P (population): humans, any age or clinical status (i.e., healthy or clinical samples), (2) I (intervention): any digital intervention for PA promotion (i.e., intervention supported by digital tools, such as smartphone apps, activity trackers, or health websites), (3) C (comparison): any other intervention or no intervention, (4) O (outcome): evaluation of any PA promotion outcome (e.g., general fitness or mobility), (5) S (study design): SR.

To reduce selection bias, SRs were independently selected by two researchers and the final consensus on inclusion was reached by discussion. Out of 304 reviews, 30 SRs met the inclusion criteria for this study. The list of included SRs is reported in our scoping review [9].

### 2.4. Procedure

To reduce rating bias, SRs were independently appraised by two researchers using AMSTAR 2 [5] and the final consensus on ratings was reached by discussion. AMSTAR 2 includes 16 items (Table 1) that can be rated “yes” (fulfilled items) or “no” (not fulfilled items). In addition, 5/16 items (items 2, 4, 7, 8 and 9) can be rated “partial yes” (if they are partially fulfilled) and 3/16 items (items 11, 12 and 15) can be rated “no meta-analysis conducted” (Table 1). The appraisal outcome is an overall confidence rating in the results of the SR that is derived based on ratings on 7/16 critical items (2, 4, 7, 9, 11, 13 and 15) and 9/16 non-critical items [5]. The confidence ratings range from “high” (no or one weakness on non-critical items), “moderate” (more than one weakness on non-critical items), “low” (one weakness on critical items) to “critically low” (more than one weakness on critical items) [5].

The appraisal procedure was performed using two approaches. Based on Approach 1, all 30 SRs were appraised with 2/16 items on AMSTAR 2 (item 2: presence of a review protocol and item 7: presence of a list of excluded studies) to identify SRs with critically low confidence ratings. Both items are critical items for deriving the overall confidence ratings (see Table 1 for the list of critical and non-critical items). These two items were often not fulfilled in SRs of health interventions [2,3,4,12] and they were selected using a fast and frugal decision tree for the critical appraisal of SRs [13]. Based on Approach 2, all 30 SRs were appraised with all 16 AMSTAR 2 items to derive the confidence ratings and to identify SR strengths and weaknesses. The appraisal outcomes (confidence ratings) for all 30 SRs were derived according to the AMSTAR 2 guidelines [5]. Item ratings were used to identify strengths and weaknesses in all 30 SRs. SR strengths were classified as fulfilled AMSTAR 2 items (i.e., items rated “yes” or “partial yes”) and SR weaknesses were classified as not fulfilled AMSTAR 2 items (i.e., items rated “no”).

### 2.5. Variables

All data were coded into a self-developed spreadsheet in Microsoft-Excel 10 (Appendix A). The coded variables included SR characteristics (first author and publication year), AMSTAR 2 appraisal outcomes (confidence ratings and item ratings) and rating time (in min/SR) for Approach 1 and Approach 2.

### 2.6. Data Analysis

All data were summarized using descriptive statistics (frequencies or means with standard deviations) and a descriptive data analysis was planned [10]. Data analysis was performed in three steps. First, the appraisal outcomes (confidence ratings and rating time) were descriptively compared between Approach 1 and Approach 2. Second, SR strengths and weaknesses were identified and descriptively summarized. Third, a sensitivity analysis was planned to compare SR strengths in SRs with better (high and moderate) confidence ratings relative to SRs with worse (low and critically low) confidence ratings.

SR strengths were expressed as percentage scores for the sensitivity analysis. According to a procedure described by others [2], item ratings for each SR were coded as 0 (“no”), 0.5 (“partial yes”) or 1 (“yes”), summed, divided by 16 items (for SRs with meta-analysis) or 13 items (for SRs without meta-analysis) and expressed as percentage scores for each SR. Mean difference scores and 95% confidence intervals (95% CI) for independent groups were used to compare SR strengths between groups. It was assumed that statistically significant difference between groups exists if the 95% CI does not include zero. All calculations were performed in Microsoft Excel 10 (Appendix A).

## 3. Results

### 3.1. SR Characteristics

The 30 SRs were published between 2007 and 2021 (Appendix A). Among all SRs, 21/30 were published after AMSTAR 2 release between 2018 and 2021 and 11/30 had review protocols.

### 3.2. Outcomes of SR Appraisal Approaches

Approach 1 (SR appraisals with two items on AMSTAR 2) was quick (mean of 5 min/SR) at identifying SRs with critically low confidence ratings (Table 2). Among all SRs, 19/30 SRs received critically low confidence ratings because they did not fulfill AMSTAR 2 item 2 (i.e., did not have a review protocol) and item 7 (i.e., did not report a list of excluded studies). Further 11/30 SRs did not receive a confidence rating, because they fulfilled one or both items 2 and 7. In this case, all 16 items on AMSTAR 2 need to be rated to derive the confidence rating.

Approach 2 (SR appraisals with 16 items on AMSTAR 2) was slower (mean of 20 min/SR) than Approach 1, but allowed to perform the full appraisals and to identify SR strengths and weaknesses (Table 2). Approach 2 showed that confidence ratings were low to critically low in 29/30 SRs and only 1/30 SRs received moderate confidence rating. There were no high confidence ratings.

### 3.3. SR Strengths and Weaknesses

Approach 2 (SR appraisals with 16 items on AMSTAR 2) was used to identify SR strengths and weaknesses in 30 SRs. Each SRs had between 0 and 13 strengths based on items rated “yes” or “partial yes” and between 2 and 11 weaknesses based on items rated “no” (Table 2). Among the weaknesses, there were between 0 and 5 critical weaknesses based on “no” ratings on critical items (Table 2).

The inspection of item ratings in all 30 SRs revealed that 9/16 items were rated “yes” or “partial yes” in most SRs (i.e., in more than 50% of 30 SRs) and 7/16 items were rated “no” in most SRs (Figure 1). Consequently, nine SR strengths based on 9/16 fulfilled items and seven SR weaknesses based on 7/16 not fulfilled items were identified.

The nine SR strengths among the 30 SRs were:Research questions and inclusion criteria were stated based on PICO (item 1);Comprehensive literature search was performed (item 4);Studies were selected in duplicate (item 5);Studies were coded in duplicate (item 6);Study details were reported (item 8);Risk of bias assessment was performed (item 9);Appropriate methods were used for meta-analysis (item 11);Heterogeneity in results was discussed (item 14);Potential sources of conflict of interest in review were reported (item 16).

The seven SR weaknesses among the 30 SRs were:Review protocol was absent (item 2);Reasons for selecting study designs were not explained (item 3);List of excluded studies was not reported (item 7);Sources of funding for primary studies were not reported (item 10);Risk of bias impact on the results of meta-analysis was not assessed (item 12)Risk of bias was not discussed (item 13).Publication bias was not assessed (item 15).

### 3.4. Sensitivity Analysis of SR Strengths

We were unable to perform the planned sensitivity analysis to compare SR strengths in SRs with better (high and moderate) confidence ratings relative to SRs with worse (low and critically low) confidence ratings due to too few SRs with better ratings (0/30 SRs with high confidence rating and 1/30 SRs with moderate confidence rating; Table 2). Instead, we performed another sensitivity analysis based on available data to compare SR strengths in SRs with review protocols relative to SRs without review protocols and in newer SRs (i.e., published after AMSTAR 2 release between 2018 and 2021) relative to older SRs (i.e., published before 2018).

More strengths (i.e., fulfilled items rated “yes” and “partial yes”) were identified in SRs with review protocols and in newer SRs (Table 3). Specifically, there were statistically significantly more SR strengths in SRs with review protocols relative to SRs without review protocols. There was also a non-significant trend toward more SR strengths in newer SRs relative to older SRs (Table 3). In addition, less critical weaknesses (i.e., critical items rated “no”) were identified in SRs with review protocols relative to SRs without review protocols, while the same number of critical weaknesses was identified in older SRs relative to newer SRs (Table 3).

## 4. Discussion

This study assessed and compared two approaches to appraising SRs of interventions in public health using AMSTAR 2. Approach 1 (appraisals with 2 items) was quick (mean of 5 min/SR) at identifying SRs with critically low confidence ratings. Approach 2 (appraisals with 16 items) was slower (mean of 20 min/SR), but allowed us to perform the full appraisals and to identify SR strengths and weaknesses. Approach 2 showed that confidence ratings were low to critically low in 29/30 SRs. More strengths were identified in SRs with review protocols relative to SRs without review protocols and in newer SRs (published after AMSTAR 2 release) relative to older SRs.

This is the first study to assess and compare different approaches to appraising SRs of health interventions using different combinations of items on AMSTAR 2. Both approaches to appraising SRs with AMSTAR 2 were useful for different purposes. The appraisal approach with two items (critical items 2 and 7) was time efficient at identifying SRs with the lowest confidence ratings, although identification of SR strengths and weaknesses was not possible using this approach. The appraisals with these two items could be performed by less experienced users of AMSTAR 2 because presence of a review protocol and a list of excluded studies can be identified relatively fast and does not require as much methodological expertise as some other items on AMSTAR 2 (e.g., item 11 that requires a judgement of methods used in a meta-analysis). Since decision makers find it difficult to select appropriate SRs for their work [14], the appraisal approach with two critical items could assist with SR classification and selection for further work. For example, such an approach can be used when large numbers of SRs on a similar topic are available for their potential application in health decision-making. In this case, a decision rule could be developed to quickly identify and exclude SRs with critically low confidence ratings from the pool of potentially relevant SRs. This can be achieved by appraising SRs with two critical items only, because critically low confidence ratings based on two critical items would not improve if all 16 items were used for appraisals. Although items 2 and 7 are often not fulfilled in SRs of health interventions [2,3,4,12], combinations of other two critical items on AMSTAR 2 could be used to quickly identify SRs with critically low ratings (see Table 1 for the list of critical and non-critical items).

While the appraisal approach with all 16 AMSTAR 2 items took longer, it allowed to perform full appraisals and to identify SR strengths and weaknesses. Our finding that most SRs of digital interventions for PA promotion have low to critically low confidence ratings has also been shown in SRs of other health interventions [2,3,4,12,15]. Two hypotheses were proposed for such poor ratings of SRs of health interventions: (1) the AMSTAR 2 tool is too conservative and tends to overestimate SR weaknesses and (2) the quality of SRs of health interventions is poor [4]. While this study was not designed to test these hypotheses, our ratings show that newer SRs published after AMSTAR 2 release tend to have more strengths than older SRs. This could be due to SR authors using AMSTAR 2 as a checklist for SR production and writing, as suggested before [16]. Furthermore, there could also be a higher awareness of the availability of reporting guidelines, such as PRISMA [17] and its newest update PRISMA2020 [18]. Despite the availability of reporting guidelines, the poor confidence ratings in this study suggest that SR authors do not adequately adhere to such guidelines. Our results also confirm the finding that SRs with review protocols have more strengths than SRs without review protocols [19,20,21,22], presumably due to better planning and preparation for SR production.

The SRs in our study had several weaknesses. Two items (a list of excluded studies, item 7, and sources of funding for primary studies, item 10) were especially poorly addressed (fulfilled in less than 10% of SRs). These results are in line with other studies [3,23]. Item 7 is particularly important for replicability of SRs and detecting any biases in study selection. Item 10 is required to assess any risk of bias in primary studies due to funding. A Cochrane review found that the results of industry-sponsored primary studies sometimes favor sponsored products, leading to more favorable efficacy results and conclusions [24]. Since the results and conclusions in SRs are based on primary studies, the information about funding should be assessed on the primary study level. Effective collaboration between industry and academic research is especially required in the field of our SRs of digital interventions for PA promotion. In addition to item 7 and 10, other weaknesses identified in this study suggest that replicability of some SRs was low and the risk of other biases was insufficiently addressed. Specifically, more than 50% of SRs in this study did not have a review protocol (item 2), did not provide reasons for the choice of study designs included in the SR (item 3), and did not assess or discuss the impact of potential sources of biases on SR outcomes (items 12, 13 and 15). Focus on the content of these items on AMSTAR 2 is required to improve the replicability of SRs and to transparently assess any potential biases that could affect SR outcomes.

Appraisal of SRs of digital interventions for PA promotion is important before such SRs can be used for practical purposes, such as policy development or health decision-making. In general, it is well known that regular PA promotes and supports both mental and physical health. However, a study that incorporated data from 358 population-based surveys in 168 countries found that the global age-standardized prevalence of inadequate PA was 27.5% in 2016 [25]. Behavior change related to PA could be supported by digital interventions involving modern technologies, such as apps or wearables [26,27]. However, it is unclear whether digital interventions to promote PA and healthy lifestyle work better alone or as a complement to in-person interventions [9] and whether they work in different populations based on age or health status [28]. There is also a need to identify factors that might increase the uptake of digital interventions for PA and improve participation in such interventions to prolong their effectiveness. The evidence addressing these issues is required from methodologically sound SRs to comprehensively and objectively assess and summarize the current state of knowledge in this rapidly developing field. AMSTAR 2 is a tool that can be used to identify such methodologically sound SRs. We show that an appraisal procedure can be shortened by first using a selection of critical items to quickly identify SRs with critical weaknesses that may not be considered for further practical use. In the second stage, all SRs without critical weaknesses on the selected critical items may be fully appraised with all 16 items on AMSTAR 2 to identify SR strengths and weaknesses. Based on such full appraisal outcomes, the SRs can be considered for further practical use.

This study had several methodological strengths. First, the risk of any biases was reduced because the study was prospectively registered [10] and two researchers selected and appraised all SRs. Second, we tested an alternative approach to SR appraisals on AMSTAR 2 and show that only two (critical) items can quickly identify SRs with the lowest confidence ratings. Third, despite poor confidence ratings in most SRs, our sensitivity analysis shows that SRs with review protocols and newer SRs tend to have more strengths. These results should encourage future SR authors to prospectively register review protocols and to adhere to reporting guidelines, including SR aspects addressed in AMSTAR 2, to improve the replicability and, thus, the overall confidence in SRs of health interventions.

There were several limitations in this study. First, we included a small sample of SRs in one field of public health. Thus, the results of this study may not be generalizable to SRs in other fields of public health and beyond. Second, we included SRs published up to 2021. This sample was selected from our scoping review [9,10] and new literature search was beyond the scope of this study. Third, due to lack of high confidence ratings, we were unable to perform a planned sensitivity analysis to compare SR strengths in SRs with better (high and moderate) confidence ratings relative to SRs with worse (low and critically low) confidence ratings. Fourth, we compared SR strengths based on two factors (presence or absence of review protocol and SR age relative to AMSTAR 2 release date). There are likely to be more predictors of SR strengths which were not analyzed in this study.

## 5. Conclusions

This study assessed and compared two approaches to appraising SRs of interventions in public health using AMSTAR 2. Only two items on AMSTAR 2 can quickly identify SRs with critical weaknesses. Although most SRs received low to critically low confidence ratings, SRs with review protocols and newer SRs tended to have more strengths. Future SRs require review protocols and better adherence to reporting guidelines to improve the confidence in their results.

## Figures and Tables

**Figure 1 ijerph-20-04689-f001:**
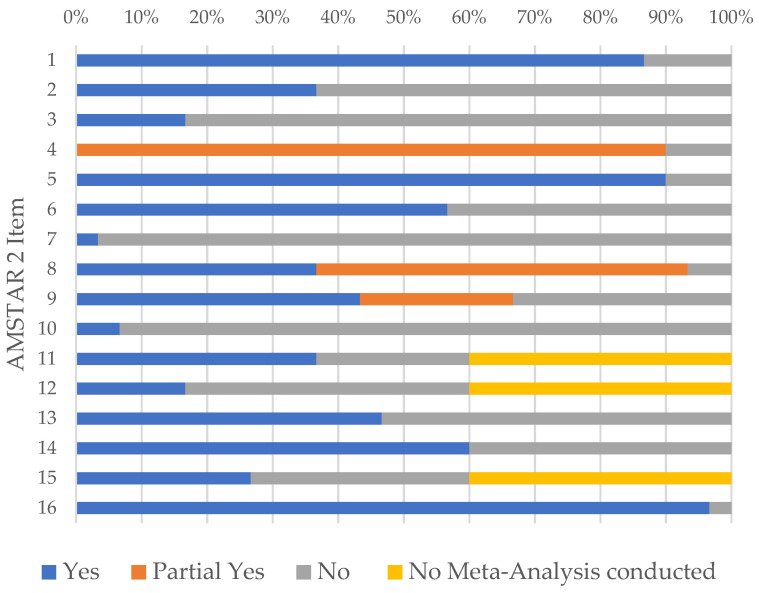
Item ratings on AMSTAR 2 in 30 SRs.

**Table 1 ijerph-20-04689-t001:** AMSTAR 2 items [5].

Item Number	Item	Rating	Critical Item
1	Did the research questions and inclusion criteria for the review include the components of PICO?	Yes No	No
2	Did the report of the review contain an explicit statement that the review methods were established prior to the conduct of the review and did the report justify any significant deviations from the protocol?	Yes Partial YesNo	Yes
3	Did the review authors explain their selection of the study designs for inclusion in the review?	Yes No	No
4	Did the review authors use a comprehensive literature search strategy?	Yes Partial YesNo	Yes
5	Did the review authors perform study selection in duplicate?	Yes No	No
6	Did the review authors perform data extraction in duplicate?	Yes No	No
7	Did the review authors provide a list of excluded studies and justify the exclusions?	Yes Partial YesNo	Yes
8	Did the review authors describe the included studies in adequate detail?	Yes Partial YesNo	No
9	Did the review authors use a satisfactory technique for assessing the risk of bias (RoB) in individual studies that were included in the review?	Yes Partial YesNo	Yes
10	Did the review authors report on the sources of funding for the studies included in the review?	Yes No	No
11	If meta-analysis was performed, did the review authors use appropriate methods for statistical combination of results?	Yes No No MA	Yes
12	If meta-analysis was performed, did the review authors assess the potential impact of RoB in individual studies on the results of the meta-analysis or other evidence synthesis?	Yes No No MA	No
13	Did the review authors account for RoB in individual studies when interpreting/discussing the results of the review?	Yes No	Yes
14	Did the review authors provide a satisfactory explanation for, and discussion of, any heterogeneity observed in the results of the review?	Yes No	No
15	If they performed quantitative synthesis, did the review authors carry out an adequate investigation of publication bias (small-study bias) and discuss its likely impact on the results of the review?	Yes No No MA	Yes
16	Did the review authors report any potential sources of conflict of interest, including any funding they received for conducting the review?	Yes No	No

Note: Items 2, 4, 7, 9, 11, 13 and 15 are critical items for deriving the confidence rating in the results of the SR. Abbreviations: AMSTAR 2, A Measurement Tool to Assess Systematic Reviews, Version 2; MA, meta-analysis; PICO, Population, Intervention, Comparison, Outcome; RoB, risk of bias; SR, systematic review.

**Table 2 ijerph-20-04689-t002:** SR appraisals with AMSTAR 2 using two approaches.

Variable	Rating	Approach 1: Appraisals with 2 Items on AMSTAR 2 (Item 2 and Item 7)	Approach 2: Appraisals with 16 Items on AMSTAR 2
		Number of appraised SRs/30	Number of appraised SRs/30
Confidence rating	high	-	0
	moderate	-	1
	low	-	2
	critically low	19	27
	none	11	-
		M ± SD (range)	M ± SD (range)
Rating time (minutes/SR)	-	5 ± 3 (2–11)	20 ± 4 (13–31)
Item rating	“yes”	-	7 ± 2 (2–13)
(number/SR)	“partial yes”	-	2 ± 1 (0–3)
	“no”	-	7 ± 2 (2–11)
	“no” (critical items)	-	3 ± 1 (0–5)

Note: Abbreviations: AMSTAR 2, A Measurement Tool to Assess Systematic Reviews, Version 2; M, mean; SD, standard deviation; SR, systematic review.

**Table 3 ijerph-20-04689-t003:** Sensitivity analysis of SR strengths based on item ratings on AMSTAR 2.

Variable	Number of SRs	Strengths (“Yes” + “Partial Yes”/All Ratings, %) M ± SD (Range)	Mean Difference [95% CI]	Critical Weaknesses (Number): M
All SRs	30	50 ± 16 (15–84)%		
Review protocol				
No	19	45 ± 15 (15–69)%		4
Yes	11	59 ± 14 (38–84)%		2
No–Yes			−14 [−25; −2]% *	
Publication date				
Older (before 2018)	9	44 ± 14 (23–62)%		3
Newer (2018–2021)	21	53 ± 16 (15–84)%		3
Older–Newer			−9 [−22; 4]%	

Note: Abbreviations: AMSTAR 2, A Measurement Tool to Assess Systematic Reviews, Version 2; CI, confidence interval; M, mean; SD, standard deviation; SR, systematic review. * statistically significant difference.

## Data Availability

All data reported in this study are shown in Appendix A.

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
