# Peer review of "Different Approaches to Appraising Systematic Reviews of Digital Interventions for Physical Activity Promotion Using AMSTAR 2 Tool: Cross-Sectional Study"

_ijerph, 2023, doi:10.3390/ijerph20064689_

Round 1

Reviewer 1 Report

Dear Authors,

Thank you for giving me the possibility to review your manuscript. I have analyzed your research with great interest. The manuscript is very well structured and has a good level of complexity. You have stated the main limitation of the study. Cited references  included mostly recent publications and relevant. The strength of  study lies in work of  two independent researchers and  consideration of different years of publication and protocol availability demonstrating that the confidence improved over time and with protocol SR.

Please, may you argue or underline that the publication is unique in the analyzed field and whether fills gap in knowledge? Is it the first study to assess different approach to an appraising SRs of interventions in public  health (or health promotion)?

Please consider the following areas of improvement:

Line 68: please, specify which type of digital intervention.  I suggest adding: “(…) any digital intervention for physical activity promotion (…)”

Line 73-74:  You stated that you identified 304 reviews. However in publications: De Santis, K.K.; Jahnel, T.; Matthias, K.; Mergenthal, L.; Al Khayyal, H.; Zeeb, H. Evaluation of Digital Interventions for 260 Physical Activity Promotion: Scoping Review. JMIR Public Health Surveill 2022, 8, e37820, doi:10.2196/37820. 261 10. De Santis, K.K.; Jahnel, T.; Mergenthal, L.; Zeeb, H.; Matthias, K. Evaluation of Digital Interventions for Physical Activity 262 Promotion: Protocol for a Scoping Review. JMIR Res Protoc 2022, 11, e35332, doi:10.2196/35332

you reported 300 reviews not 304. Please check and clarify.

Line 139: Lack of number 0 before “-11 weakness (…)”

Table 1: Line: “Confidence rating: requires 16 items” maybe add after items “assessment”

Table 1: I suggest adding in empty fields “-“ or “not applicable  (NA)”

Table 1:  I suggest adding abbreviations: M – mean, SD – standard deviation

3.3. Sensitivity Analysis: Whether difference between with vs. without review protocols and in newer vs. older SRs is statistically significant? Consider to use statistical methods to estimate significant.

3.3. Sensitivity Analysis: Please, consider to add information about lack of comparison strengths in SRs with critically low and low vs. moderate and high confidence ratings. My suggestion is:

In the current analysis the number of SRs assessed as high and moderate confidence rating was too small (0 with high and 1 with moderate rate) to conduct meaningful sensitivity analyses to compare strengths in SRs with critically low and low vs. moderate and high confidence ratings.

4. Discussion Second paragraph: Analysis shown that protocol preparation and date of publication are two predictors of better quality SRs. It is very important result! Maybe you can consider to add that this predictors are important but not the only ones and it can mbe more potential predictors of high quality SRs which were not analyzed in this study.

Line 216-218: You stated: “Two items on AMSTAR 2 can quickly identify SRs that should not be considered for evidence-based decisions in public health.” In my opinion short assessment (2 items) cannot replaced full assessment which is comprehensive, but maybe you consider to add that Approach 1 is good first step to exclude critically low quality SRs and the remaining SRs with uncertain assessment (fulfilled one of both items 2 and 7) should be comprehensively assess.

Best wishes,

Reviewer 2 Report

Thank you for the opportunity to review this paper. Please see attachment for my comments and suggestions. 
